# Fast Way to Predict Parking Lots Availability: For Shared Parking Lots Based on Dynamic Parking Fee System

Sheng-Ming Wang [1] and Wei-Min Cheng [2,*]

1  Department of Interaction Design, National Taipei University of Technology, Taipei 10608, Taiwan
2  Doctoral Program in Design, College of Design, National Taipei University of Technology, Taipei 10608, Taiwan
*  Correspondence: t108859003@ntut.edu.tw

**Abstract:** This study mainly focuses on the estimation calculation of urban parking space. Urban parking has always been a problem that plagues governments worldwide. Due to limited parking space, if the parking space is not controlled correctly, with the city's development, the city will eventually face the result that there is nowhere to park. In order to effectively manage the urban parking problem, using the dynamic parking fee pricing mechanism combined with the concept of shared parking is an excellent way to alleviate the parking problem, but how to quickly estimate the total number of available parking spaces in the area is a big problem. This study provides a fast parking space estimation method and verifies the feasibility of this estimation method through actual data from various types of fields. This study also comprehensively discusses the changing characteristics of parking space data in multiple areas and possible data anomalies and studies and explains the causes of data anomalies. The study also concludes with a description of potential applications of the predictive model in conjunction with subsequent dynamic parking pricing mechanisms and self-driving systems.

**Keywords:** parking lots; parking availability; parking prediction; parking fees; shared parking

## 1. Introduction

Due to the evolution of the times and the improvement of technology, it is no longer difficult for the public to purchase vehicles. Some people even have multiple cars in their homes. However, the number of vehicles that grows yearly has caused other social problems. Due to the high cost of new car parks and rising land prices in urban areas, governments cannot provide enough parking space for the growing number of vehicles. The urban parking demand is often concentrated in the city center or popular locations. However, in the city center, where land is expensive, in addition to the fact that the land price is much higher than that of the general area, it is impossible to use all the land for parking space. Therefore, the city's parking problem is becoming more serious. More seriously, when many vehicles that cannot find a parking space are constantly traveling back and forth on the city's roads to find a parking space, the urban traffic is often paralyzed, and more follow-up traffic problems are caused. Faced with such a parking dilemma, governments worldwide have carried out a lot of research and tried to find various feasible policies or solutions. These studies include: pre-preparing parking spaces by predicting supply and demand [1], or promoting shared parking spaces [2], and reducing vehicle traffic through price or other means [3].

Governments have tried to encourage citizens to take public transport actively. However, due to the difficulty of changing their habits, the general public is still accustomed to driving their vehicles to their destinations. As a result, not only are traffic jams caused, but even parking spaces are hard to find. Urban parking spaces are often close to the average usage level only at night. Once the public starts to work during the day, all suburban

vehicles will be concentrated in the city center, which will cause great trouble to park in the city center. For another example, during consecutive holidays, due to the influence of tourism, the traffic flow may move across counties and cities on a large scale. Therefore, the total number of parking spaces a specific county or city location requires may be much higher than the total number of parking spaces that the county and city location usually has. If the parking problem cannot be fully resolved, not only will people's interest in traveling greatly diminish, but it will also have a particular impact on the tourism market. Therefore, whether to release all the available urban parking spaces as much as possible has become a critical topic, and we also found that some studies have made efforts to investigate this. For example, providing parking guidance [4,5] through prediction technology to help drivers find parking spaces on the side of the road more quickly, or improving prediction algorithms based on sensor data to evacuate real-time traffic flow faster [6,7]. Recently, with the evolution of the neural network model, some new predictive models that mix the first two features and can handle more complex data have also appeared [8,9].

In order to release more parking spaces, the concept of shared parking spaces has also occurred. In order to accurately estimate the available parking spaces in the shared parking space mode, it is necessary to improve the parking space prediction method. First, some researchers studied how to optimally allocate these parking spaces [10], and then tried to make further improvements from the aspects of parking preferences and behaviors [11], and managed to release more parking spaces. As the problem of arrears leads to supply and demand problems, some scholars have also studied methods to reduce delayed payments [12].

The original purpose of sharing parking spaces is to provide more parking spaces for vehicles by recruiting idle, scattered parking spaces in the city, thereby alleviating the problem of urban parking. Although this approach can provide some parking spaces to facilitate urban parking, the number is limited, and it still cannot fully fulfill the parking demand during peak urban traffic hours. To find out the available parking spaces in the city, the shared parking spaces are not only expanded from the standard parking lots of government units, community houses, or government public houses, but also private enterprises or private homes.

The considerations for the release of different types of shared parking spaces may be different. Some studies have found that owners of private parking spaces may be more concerned about benefits [13], but citizens of different scales may also have different considerations [14]. For residential areas, there may be other management considerations [15]. Some studies have also shown that the shared parking space strategy has certain risks [16]. This strategy does not always guarantee that the site owners can profit smoothly but may make them compete with traditional parking lot operators, so the effect of different locations is not the same. However, it is worth noting that, compared with considerable costs to build parking lots, for governments around the world, actively promoting and advocating the use of shared parking spaces is still a more feasible solution to the urban parking problem.

Although the government actively encourages the use of public transportation and reduces the cost of public transit, the habit of people driving by themselves is still difficult to change, and excessive restraint on driving will also harm the auto industry. Under the consideration of multi-party policies, the parking problem will gradually be handled in the direction of price-based quantity.

Since the parking lot currently only provides parking services based on a fixed parking fee per hour, obviously, all drivers will grab a parking space on a first-come-first-served basis. As long as one parks in a parking space, because the hourly fee is fixed and low, one will not cherish the parking space, and it is more likely that one will park for a few hours or a day. Prolonged stopping is the main culprit behind the low efficiency of urban public parking spaces. Especially in the city center or places with dense parking demand, if the parked vehicles cannot leave early, it will be a big problem for more drivers who need to park but have no space to park.

Recently, studies have pointed out that the parking problem may be improved by using different pricing strategies [17]. To improve the situation of long-term parking, some parking lots have adopted a progressive rate method to increase the parking fee as the parking time becomes longer. Such an adjustment will indeed improve the long-term parking of vehicles and will also release more parking spaces in each parking lot to some extent. However, for the city center with intensive parking demand, it is still impossible to change the habit of the public who like to drive by themselves. If there is no parking space in a specific parking lot, the driver only needs to find a nearby parking lot with vacancies to park, which is also a behavior shown in general driving.

If all parking spaces in the adjacent area can be priced according to the current supply and demand in the market, the situation may be different [18]. That is, the hourly parking fee should not be fixed. It can be cheap when no vehicles are parked and expensive when many cars rush for a space. The parking lot should constantly announce the latest hourly parking space number and let drivers know the possible parking fee at a certain point in the future so that they can judge whether to leave early. In addition, it can also be used as a reference index for pricing individual driving needs according to the actual conditions related to the car owner, such as the driving destination, the number of people in the car, and the walking time [19].

To do this, one must first be able to quickly grasp the supply and demand of all adjacent parking spaces in the area. That is to say, the current number of available parking spaces and the possible number of parking spaces in the future for each parking lot in the adjacent area must be determined, and then the subsequent dynamic supply and demand pricing design can be carried out. The purpose of this research is to find a method that can estimate the number of available parking spaces at a particular time in the future through the parking records of the parking lot so that the system can be used in subsequent pricing strategies. The focus of this study is how to quickly estimate the total number of available parking spaces in all adjacent parking lots in the most efficient and general way.

The sections of this research are arranged as follows: Section 2 includes a review of some modeling methods and field case studies. Section 3 describes the model establishment and verification methods. Section 4 describes the results of the field demonstration and Section 5 discusses some special cases and data processing issues. Section 6 summarizes and describes future development directions.

## 2. The Current Study

In order to predict the available parking spaces more accurately, various advanced prediction methods have been proposed, such as deep learning-based CNN-LSTM [20], transfer learning framework [21], the bagging regression algorithm [22], random forest [23], the autoregressive moving average model ARIMA [24], voting-based hybrid ensemble classification [25], continuous-time Markov chain [26], hybrid space–time graph volume product network (HST-GCN) [27], and other methods. The various models described above can be used to estimate parking space occupancy rates, and they seem to have reached a good level of prediction. Although these models analyze factors such as weather, events, tolls, regions, time, etc., and establish some data-driven models, there are still some significant problems in these practices:

- Learning through data can provide a model that fits the data, but this only means that the model is close to the data behavior to a certain extent, it does not mean that the model is explanatory, and it does not even guarantee that it is correct. Even if we put together the parking data of two different morphologies, day and night, we can still obtain a model. However, is such a model really suitable for our use?
- Due to the field's characteristics, some parking lots' parking patterns may be regular, while the parking patterns of some parking lots may be irregular. Since we cannot predict the possible parking patterns of the parking lot in advance, even if it is expected to be a particular parking pattern at the beginning, this parking pattern may still change in the future. Using machine learning, however, is vulnerable to the shock of

such a shift. Abnormal data or dirty data often make the model fit more and more biased.

- When a parking lot is established in practical application scenarios, the prediction system usually needs to be used immediately. That is to say, the actual situation is that we cannot collect enough data for the system to learn, or there are no so-called historical data to learn. Therefore, it is necessary to try to reduce the amount of learning data required by the system, and at the same time, it can meet various types of field predictions.

Below, we use some real case studies to observe the actual parking conditions of various types of parking lots. We made some observations on the situation of some parking lots through the open data of the Taipei City Government's parking information and obtained some comments:

### Commercial office type locations (fixed entry and exit period)

Most of the parking lots belong to this type, mainly based on the parking needs of general office workers. They start to park during the morning work hours and gradually leave when they leave work in the evening, with no vehicles coming to park on weekends or holidays. The parking situation of this type of parking lot is the best estimate of all kinds, and the changes are relatively small.

### Mall type locations (people are coming and going at any time)

These kinds of locations typically include department stores. Whether it is a weekday or a holiday, visitors are coming in and out at any time from morning to night, so the parking situation of the parking lot will also change correspondingly with the number of visitors and the average stay time of visitors. When businesses close for the day, the parking lot is usually back to high availability because all visitors have left.

### Exhibition hall type locations (there will be visitors on unspecified dates)

Similar to the first type of commercial office locations, parking conditions are mainly divided into two types: exhibition days and non-exhibition days. As long as the type of day can be distinguished, the parking situation performance can be estimated directly according to the type.

### Special business district type locations (some periods are prone to drastic changes)

Similar to the second type of mall location, this is also a type of location where visitors come and go throughout the day. Since the business district is composed of various industries, the vehicle parking behavior is also diverse. As a result, there may be many irregular entries and exits in some periods for various reasons.

### Other locations (always available, infrequent in and out)

Here, we also noticed an interesting phenomenon. Although some places with high demand for parking are often difficult to find available spaces, some parking lots in the same area are often left with no drivers willing to park. These car parks include hospitals, traditional markets, and nearby funeral homes. Perhaps due to different customs in different countries, drivers may be more reluctant to go to the parking lot of some locations, so the parking spaces in those places are often vacant and underutilized.

After observing some parking lots' available parking space data, we also found some data change characteristics. Generally speaking, we use the same periodic data as the inference basis for possible future results, such as using the available parking spaces at 8 p.m. on several consecutive Fridays to estimate the parking spaces at 8 p.m. on a specific Friday in the future. This method is acceptable if used in a parking lot with regular cycles. Still, if it is a holiday or used in an irregular parking lot, such a prediction method may cause significant errors. For example, in the exhibition type field, due to the visitors' date uncertainty, the available parking spaces in the parking lot may be completely different even in the same cycle time.

Compared with the average value of the same period, the more critical reference value may be the value of the two hours before and after. Through the observation of the above case, it can be found that the value of each period is related to the value of the previous two periods. Generally speaking, the current period's value can be inferred by the ratio of the previous two periods. However, some periods sometimes have a lot of vehicles moving in and out, so there will be no small changes in the period's value. In this case, the change ratio of the previous two periods cannot predict the period's value. Even so, for the same period on different days, the fluctuation ratio of the change ratio is not significant. For example, suppose the number of available parking spaces is reduced by 20% at 8 o'clock yesterday compared to 7 o'clock yesterday. In that case, the number of available parking spaces at 8 o'clock today should be reduced by about 20% compared with 7 o'clock today. This ratio (20%) has nothing to do with the total number of parking spaces on the day and is often a fixed value.

The above case study results give us some inspiration. We also found that some past studies have made similar points, such as: parking classification is necessary for algorithm evaluation use [28,29], the parking conditions on weekdays and holidays are different and need to be handled separately [30,31], the numbers of parking spaces in the previous hour and the next hour are highly correlated [32,33] etc. With these observations and experiences as a basis, we try to build appropriate models and evaluate whether the models we produce are suitable for use in the actual field of the various situations above.

## 3. Modeling Method

Taipei is one of the world's most famous cities, which contains a variety of commercial activity areas, and various commercial activities are frequently carried out 24 h a day. Therefore, this city as a sampling reference for this study is sufficient to cover most possible parking scenarios in Asia.

In Taipei, are at least 1000 parking lots. It obviously requires a lot of computing resources to retrieve the available parking spaces from each parking lot and then use a neural network for modeling and estimation. Therefore, such an approach struggles to achieve the goal of a city-scale dynamic pricing system for regional parking fees. Therefore, to quickly process the estimation of the number of available parking spaces of many parking lots in a specific area, we use a fast and efficient rule-based modeling method.

This study does not focus on the changes in parking conditions in the time series but instead explores its characteristic points. Although the prediction is also based on big data, compared with the neural network operation using deep learning, this research adopts a statistical verification method and cooperates with effective operation rules. This is so that data utilization is no longer a black-box operation, and it can also reduce the influence of data overfitting.

After reviewing various estimation algorithms, we know that the number of available parking spaces in a parking lot is often not changed by linear averaging. After careful examination, the following vital rules for modeling can be obtained:

**Important Rule 1:**

Even in the same parking lot, the number of available parking spaces in the same period often differs from day to day. However, the number of available parking spaces in the two hours before and after is often very similar. Therefore, if the numbers of available parking spaces for three consecutive hours are respectively called "number of available parking spaces_current," "number of available parking spaces_before one hour," and "number of available parking spaces_before two hours," then the three should have the following relationship:

$$\text{Available}_{\text{before two hours}} : \text{Available}_{\text{before one hour}} = \text{Available}_{\text{before one hour}} : \text{Available}_{\text{current}}$$

Thus,

$$\text{Available }_{current} = \text{Available }_{before\ one\ hour} * \frac{\text{Available }_{before\ one\ hour}}{\text{Available }_{before\ two\ hours}}$$

**Important Rule 2:**

If the current hour happens to be a working time or a popular time, obviously, the number at this time will not change with the ratio of the previous two hours, and the number at the same time of each day will not be the same. However, the rate of change between this time of day and the previous hour is often still the same. As long as the average daily rate of change of the field for this period is obtained, it can be estimated based on the number of the previous hour:
Thus,

$$\text{Available }_{current} = \text{Available }_{before\ one\ hour} * Avg\left(\frac{\text{Available }_{current}}{\text{Available }_{before\ one\ hour}}\right)$$

If one wants to estimate for any period of the day, as long as one can obtain the historical data for the first two consecutive hours from that period, one can calculate for that period. The "AVG" referred to here is just a very simple average. For example, if we have the ratio of available parking spaces for a specific period in the past month, we can simply average the ratio of each day in this month. This average value is usually not affected by the conditions of the day and is always fixed. This study also uses this important feature to make predictions.

However, there are still two issues that need to be addressed.

First, as mentioned in Important Rule 2 above, we know that some periods have different ratios of change. Since the calculation formula of the period is different, how do we know whether the period to be calculated belongs to the standard period or the change period?

Second, we know that the performance of the general field on weekdays and holidays is not the same, and the weekdays of some particular fields are also different from those of the general field. Days with significant changes will seriously affect the calculated average. Therefore, before calculating the norm, it is necessary to find out the weekdays and holidays of each field and calculate them separately, but how do we find out the weekdays and holidays of the field?

**Important Rule 3:**

We know that there are 24 h in a day. Suppose all consecutive hour numbers are subtracted in pairs, and the numbers obtained in sequence are defined as $\text{DiffH}_0$, $\text{DiffH}_1$, $\text{DiffH}_2$, ... , $\text{DiffH}_{22}$, $\text{DiffH}_{23}$. The period with a massive change in the day will correspond to those values with higher absolute values among the 24 numbers (as shown in Figure 1). Whether it is more or less, it indicates abnormal changes. Therefore, by finding the extreme values in these numbers, one can obtain the abnormal change period in the day. If these periods are encountered during modeling calculations, the formulas mentioned in Important Rule 2 should be used instead.

|  | $h_0$ | $h_1$ | $h_2$ | $h_3$ | $h_4$ | $h_5$ | $h_6$ | $h_7$ | $h_8$ | $h_9$ | $h_{10}$ | $h_{11}$ | $h_{12}$ | $h_{13}$ | $h_{14}$ | $h_{15}$ | $h_{16}$ | $h_{17}$ | $h_{18}$ | $h_{19}$ | $h_{20}$ | $h_{21}$ | $h_{22}$ | $h_{23}$ |
|---|---|---|---|---|---|---|---|---|---|---|---|---|---|---|---|---|---|---|---|---|---|---|---|---|
| diff | 0 | 0 | 0 | 0 | 0 | 0 | 0 | 0 | 6 | 1 | 0 | 0 | 3 | 1 | 0 | 0 | 1 | 6 | 3 | 1 | 0 | 0 | 0 | 0 |
| average | 1 | 1 | 1 | 1 | 1 | 1 | 1 | 1 | 1 | 1 | 1 | 1 | 1 | 1 | 1 | 1 | 1 | 1 | 1 | 1 | 1 | 1 | 1 | 1 |
| z-score | 0 | 0 | 0 | 0 | 0 | 0 | 0 | 0 | 2 | 0 | 0 | 0 | 1 | 0 | 0 | 0 | 0 | 2 | 1 | 0 | 0 | 0 | 0 | 0 |

**Figure 1.** Demonstration of the diff dataset.

**Important Rule 4:**

The last key is how to find the weekdays and holidays. In Important Rule 2, the concept of the average amount of change in each period of the day was mentioned. We know that the performance during certain holiday periods with no changes is not much different

from that of weekdays. However, as long as there is a popular period, the performance of the holiday period will definitely be significantly different from the performance of the weekday period. Generally speaking, the total difference of holidays will be much higher than the total difference of weekdays, so the result after the average calculation is bound to be similar to the performance on weekdays. That is to say, if we compare the data performance of each holiday period on any day with the average value, the total difference accumulated during the 24 h holiday is bound to be much higher than the total difference calculated on weekdays (as shown in Figure 2). It is to use such characteristics to judge whether any day is a weekday or a holiday. In fact, this is a chi-square goodness-of-fit test, mainly through comparing the hourly average of a specific day and a normal day, to identify the days judged as unsuitable as holidays.

|  | $h_0$ | $h_1$ | $h_2$ | $h_3$ | $h_4$ | $h_5$ | $h_6$ | $h_7$ | $h_8$ | $h_9$ | $h_{10}$ | $h_{11}$ | $h_{12}$ | $h_{13}$ | $h_{14}$ | $h_{15}$ | $h_{16}$ | $h_{17}$ | $h_{18}$ | $h_{19}$ | $h_{20}$ | $h_{21}$ | $h_{22}$ | $h_{23}$ |  |
|---|---|---|---|---|---|---|---|---|---|---|---|---|---|---|---|---|---|---|---|---|---|---|---|---|---|
| Normal Day | 0 | 0 | 0 | 0 | 0 | 0 | 0 | 0 | 3 | 1 | 1 | 1 | 2 | 1 | 1 | 1 | 1 | 3 | 1 | 0 | 0 | 0 | 0 | 0 |  |
| Daily average | 0 | 0 | 0 | 0 | 0 | 0 | 0 | 0 | 3 | 1 | 1 | 1 | 2 | 1 | 1 | 1 | 1 | 3 | 1 | 0 | 0 | 0 | 0 | 0 |  |
| diff | 0 | 0 | 0 | 0 | 0 | 0 | 0 | 0 | 0 | 0 | 0 | 0 | 0 | 0 | 0 | 0 | 0 | 0 | 0 | 0 | 0 | 0 | 0 | 0 | 0 |
|  |  |  |  |  |  |  |  |  |  |  |  |  |  |  |  |  |  |  |  |  |  |  |  |  |  |
| Holiday | 1 | 0 | 0 | 0 | 0 | 0 | 0 | 0 | 0 | 0 | 0 | 0 | 3 | 0 | 0 | 0 | 0 | 0 | 2 | 2 | 2 | 2 | 2 | 2 |  |
| Daily average | 0 | 0 | 0 | 0 | 0 | 0 | 0 | 0 | 3 | 1 | 1 | 1 | 2 | 1 | 1 | 1 | 1 | 3 | 1 | 0 | 0 | 0 | 0 | 0 |  |
| diff | 1 | 0 | 0 | 0 | 0 | 0 | 0 | 0 | 3 | 1 | 1 | 1 | 1 | 1 | 1 | 1 | 3 | 1 | 2 | 2 | 2 | 2 | 2 | | 26 |

**Figure 2.** Demonstration of the diff dataset on weekday/holiday.

Through the above calculation methods, we can quickly use a small amount of historical data for modeling, and quickly calculate the estimated amount of the current hour. For data collection or statistical analysis methods, one can refer to general data analysis practices. The method provided in this study is mainly used for forecasting applications rather than data collection. Please refer to Figure 3 below for the actual application scenarios of each rule.

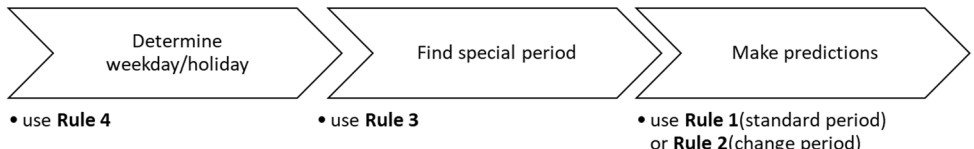

**Figure 3.** The actual application scenarios of each rule.

Next, we try to verify the accuracy of our hypothetical model with real-world parking lot data. We use linear regression to examine the degree of correlation between actual and predicted values.

## 4. Results

In order to verify whether our proposed prediction method is feasible, we use the data modeling method introduced in the previous section for data prediction, and verify our model through the data of many real-world parking lots in a large city.

Sampling method:

From the open data of parking information in Taipei City (Retrieved 10 November 2021, from https://data.taipei/), two random samplings of weekday data of about three weeks were selected (as shown in Table 1), and 80 parking lots with more than 300 parking spaces were selected as observation objects. The parking lot locations can be seen in Figure 4 below. The chosen parking lot locations are evenly distributed in the city and are of various types and backgrounds. There is no problem with sampling bias.

**Table 1.** Start date and end date of sampling.

|  | Start Date | End Date |
|---|---|---|
| First sampling | 10 November 2021 | 6 December 2021 |
| Second sampling | 16 December 2021 | 11 January 2022 |

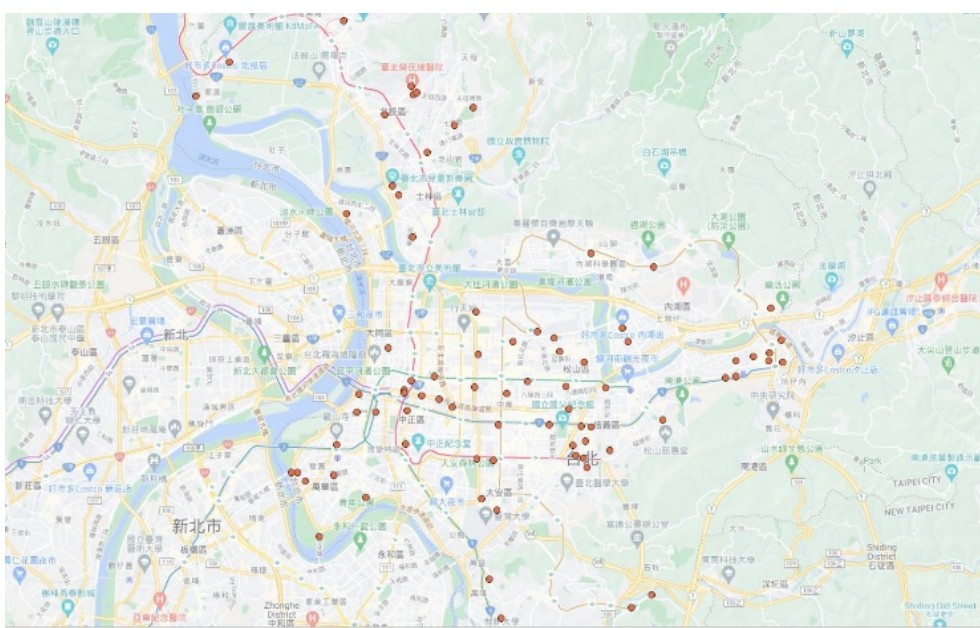

**Figure 4.** Overview of parking lot distribution.

Data schema:

The parking lot information fields we mainly collect include the following five items (as shown in Table 2):

**Table 2.** Explanation of data field of parking lots.

| Field Name | Field Description |
|:---:|:---:|
| id | Identification number of the parking lot |
| name | The name of the parking lot |
| totalcar | The total number of available parking spaces in the parking lot |
| tw97x | Location information of the parking lot (TWD97 X coordinate value. TWD97 is a global coordinate system dedicated to Taiwan) |
| tw97y | Location information of the parking lot (TWD97 Y coordinate value. TWD97 is a global coordinate system dedicated to Taiwan) |

The parking record fields we mainly collect include the following four items (as shown in Table 3):

**Table 3.** Explanation of data field of parking records.

| Field Name | Field Description |
|:---:|:---:|
| id | Identification number of the parking lot |
| date | Parking date |
| hour | Current time (sampled every hour) |
| availablecar | Number of parking spaces available |

Results of regression analysis:

We use regression analysis to see how closely the actual and predicted values are correlated. Among the 80 cases of parking lots that were counted, 64 records have an $R^2$ of more than 0.7 and 10 have an $R^2$ of less than 0.5. Overall, the estimated model is valid for more than 80% of the fields, and 42% have $R^2$ above 0.9. The $R^2$ value of each field can be seen in Table 4 below, and the distribution pattern can be seen in the thumbnail (as shown

in Figure 5). It can be found that in most cases, the predicted value is close to the actual value, so the points on the graph move closer to the regression line. However, there are still a few cases where the graph exhibits anomalous scatter.

Compared to other modeling methods:

We also use a neural network to model and make predictions. Using the above dataset, after trying to model, it is found that because the results of each sampling are different, the result of the prediction model obtained after each training is also different. We train each case 20 times and observe the maximum, minimum, and average of the predicted correlations. After observation, it is found that the prediction accuracy fluctuates randomly between about 10% and 70% (as shown in Table 5).

**Table 4.** The correlations between the actual and predicted values.

| id | $R^2$ | id | $R^2$ | id | $R^2$ | id | $R^2$ |
|---|---|---|---|---|---|---|---|
| 1 | 95.31% | 56 | 44.53% | 165 | 92.85% | 376 | 91.05% |
| 2 | 93.50% | 58 | 95.84% | 173 | 93.02% | 377 | 16.27% |
| 7 | 95.15% | 59 | 65.01% | 174 | 80.81% | 378 | 92.23% |
| 14 | 87.15% | 67 | 93.97% | 183 | 85.93% | 379 | 93.07% |
| 15 | 25.68% | 73 | 54.41% | 184 | 67.04% | 381 | 83.90% |
| 16 | 94.39% | 75 | 70.98% | 194 | 92.00% | 384 | 62.77% |
| 18 | 70.96% | 78 | 97.74% | 203 | 87.64% | 421 | 45.59% |
| 21 | 75.95% | 79 | 95.55% | 214 | 92.90% | 462 | 90.00% |
| 23 | 97.36% | 80 | 68.30% | 217 | 78.22% | 472 | 89.63% |
| 27 | 94.44% | 85 | 81.57% | 234 | 92.31% | 483 | 97.86% |
| 29 | 77.27% | 87 | 75.01% | 255 | 54.84% | 487 | 97.74% |
| 32 | 89.76% | 88 | 64.60% | 260 | 97.11% | 570 | 96.62% |
| 37 | 87.27% | 91 | 93.81% | 299 | 11.15% | 585 | 85.60% |
| 41 | 88.73% | 96 | 56.64% | 301 | 96.60% | 593 | 98.25% |
| 42 | 92.81% | 100 | 13.80% | 338 | 89.03% | 595 | 62.51% |
| 43 | 84.66% | 101 | 79.79% | 340 | 85.88% | 608 | 90.37% |
| 44 | 87.52% | 103 | 91.84% | 343 | 76.64% | 614 | 85.78% |
| 48 | 93.74% | 115 | 52.76% | 361 | 84.16% | 631 | 82.46% |
| 50 | 92.50% | 120 | 94.21% | 365 | 94.73% | 639 | 81.69% |
| 53 | 93.47% | 124 | 94.35% | 372 | 80.74% | 650 | 74.41% |

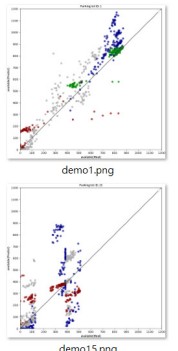

demo1.png

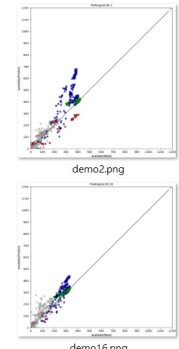

demo2.png

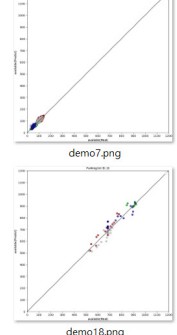

demo7.png

demo14.png

demo15.png

demo16.png

demo18.png

demo21.png

**Figure 5.** *Cont.*

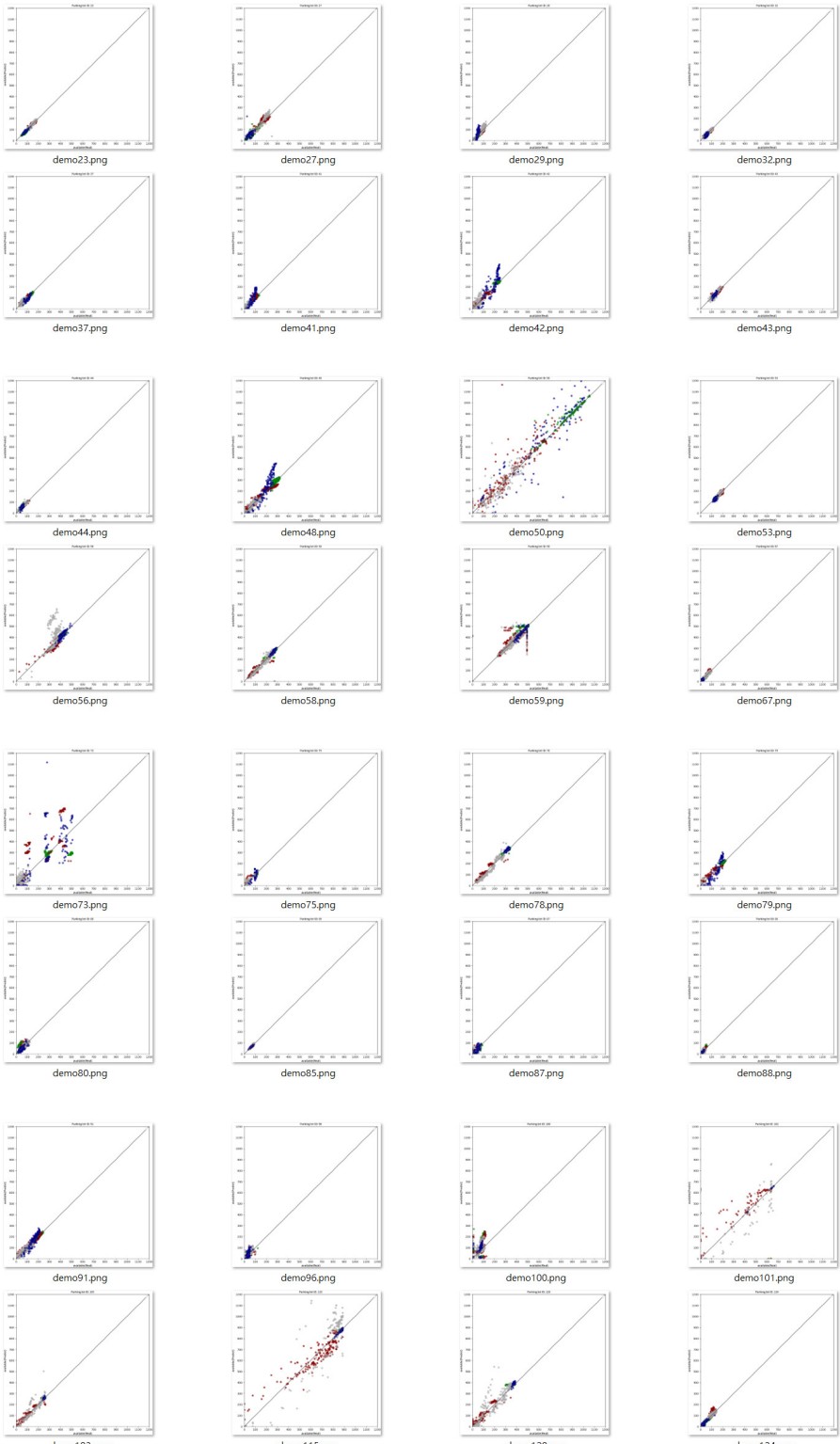

**Figure 5.** *Cont.*

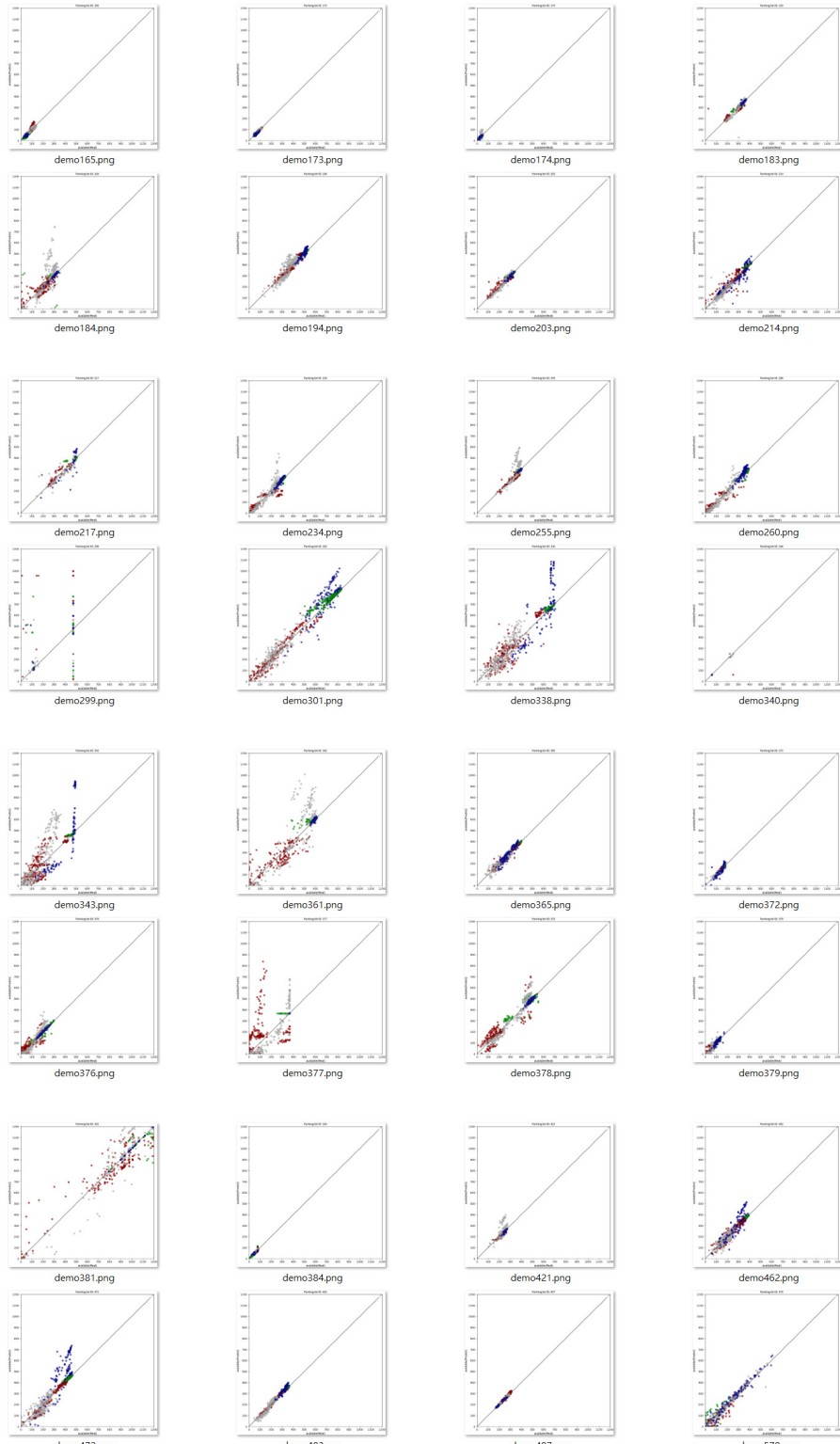

**Figure 5.** *Cont.*

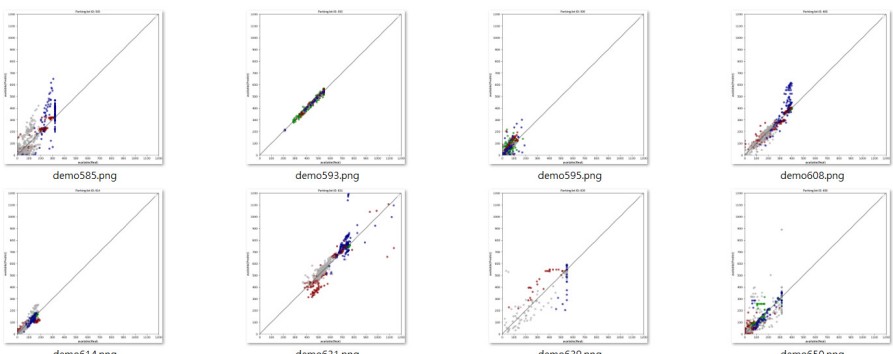

**Figure 5.** The distribution pattern of all 80 cases.

**Table 5.** Correlation trained with neural networks.

| id | Max R | Min R | Mean R | id | Max R | Min R | Mean R |
|---|---|---|---|---|---|---|---|
| 1 | 32.60% | 25.17% | 29.29% | 165 | 43.79% | 42.64% | 43.16% |
| 2 | 53.02% | 48.77% | 51.44% | 173 | 36.36% | 33.40% | 35.47% |
| 7 | 29.39% | 27.11% | 28.80% | 174 | 37.80% | 18.06% | 24.85% |
| 14 | 21.72% | 13.07% | 18.59% | 183 | – | – | – |
| 15 | 49.36% | 44.28% | 47.16% | 184 | 22.26% | 14.95% | 17.68% |
| 16 | 46.16% | 44.44% | 45.17% | 194 | 46.85% | 46.30% | 46.55% |
| 18 | 28.53% | 28.05% | 28.33% | 203 | 23.32% | 20.23% | 22.10% |
| 21 | 10.31% | 5.95% | 7.65% | 214 | 47.75% | 39.29% | 46.28% |
| 23 | 55.03% | 53.96% | 54.51% | 217 | – | – | – |
| 27 | 32.75% | 31.70% | 32.32% | 234 | 27.35% | 15.11% | 22.94% |
| 29 | 9.89% | 6.62% | 8.64% | 255 | 17.47% | 14.76% | 16.30% |
| 32 | 24.21% | 16.63% | 20.97% | 260 | 36.86% | 34.71% | 36.14% |
| 37 | 43.39% | 42.32% | 42.96% | 299 | 20.94% | 17.67% | 19.75% |
| 41 | 58.77% | 55.11% | 57.02% | 301 | 51.40% | 47.74% | 49.92% |
| 42 | 56.75% | 54.10% | 55.68% | 338 | 50.40% | 47.43% | 48.74% |
| 43 | 6.66% | 0.76% | 3.70% | 340 | 7.82% | 0.24% | 3.51% |
| 44 | 28.57% | 21.13% | 25.38% | 343 | 45.47% | 42.19% | 44.40% |
| 48 | 54.06% | 42.34% | 50.50% | 361 | 17.85% | 13.30% | 16.35% |
| 50 | 60.34% | 57.00% | 58.50% | 365 | 61.44% | 58.38% | 59.56% |
| 53 | 30.01% | 26.20% | 28.46% | 372 | 76.44% | 74.71% | 75.53% |
| 56 | 40.79% | 36.99% | 40.10% | 376 | 12.10% | 5.71% | 9.56% |
| 58 | 27.42% | 25.02% | 25.89% | 377 | 9.37% | 0.02% | 4.26% |
| 59 | 20.92% | 16.63% | 18.28% | 378 | 21.31% | 20.24% | 20.89% |
| 67 | 21.83% | 20.47% | 21.38% | 379 | 41.83% | 38.38% | 40.41% |
| 73 | 38.56% | 36.60% | 37.56% | 381 | 1.35% | 0.23% | 0.61% |
| 75 | 41.23% | 38.55% | 40.01% | 384 | – | – | – |
| 78 | 32.59% | 30.07% | 31.64% | 421 | 34.03% | 30.58% | 32.81% |
| 79 | 58.59% | 47.56% | 56.25% | 462 | 71.97% | 69.06% | 71.14% |
| 80 | 36.59% | 35.67% | 36.12% | 472 | 60.85% | 59.63% | 60.23% |
| 85 | 42.52% | 41.85% | 42.20% | 483 | 53.89% | 52.39% | 53.39% |
| 87 | 39.72% | 11.20% | 28.91% | 487 | 7.07% | 3.52% | 5.92% |
| 88 | 6.81% | 0.88% | 4.34% | 570 | – | – | – |
| 91 | 67.04% | 63.41% | 66.12% | 585 | 51.67% | 47.49% | 49.94% |
| 96 | 4.82% | 3.02% | 3.72% | 593 | 41.91% | 37.34% | 39.10% |
| 100 | 4.34% | 2.53% | 1.96% | 595 | 31.38% | 24.81% | 28.95% |
| 101 | 24.43% | 16.85% | 21.07% | 608 | 58.23% | 56.92% | 57.54% |
| 103 | 20.68% | 19.52% | 20.07% | 614 | 23.13% | 14.20% | 19.01% |
| 115 | 24.09% | 22.15% | 23.35% | 631 | 54.67% | 54.11% | 54.36% |
| 120 | 25.75% | 24.11% | 24.92% | 639 | 33.67% | 28.10% | 30.58% |
| 124 | 42.73% | 41.42% | 42.14% | 650 | 12.10% | 10.41% | 11.40% |

## 5. Discussion

In most cases, good prediction results can be obtained using the fast model estimation in this study. The prediction accuracy may be higher or lower in a few exceptional cases. We can obtain some inspiration by observing and analyzing some cases. According to different types of prediction result distribution diagrams, they can be roughly divided into four categories, namely "close fit," "straight line," "horizontal line," and "mass of blocks."

Close fit:

Usually, such cases are essential parking lots that government units pay more attention to, which may be located in vital business districts (case 78, Figure 6), stations (case 483, Figure 7), markets (cases 593 and 487, Figures 8 and 9), etc., and are accompanied by good management or unlimited daily charges. Due to this, these parking lots are less likely to experience abnormal communication or long-term parking of vehicles, and the variability of each period during the day is also moderate. As the changes align with the usual way of change, more accurate forecast results can be obtained.

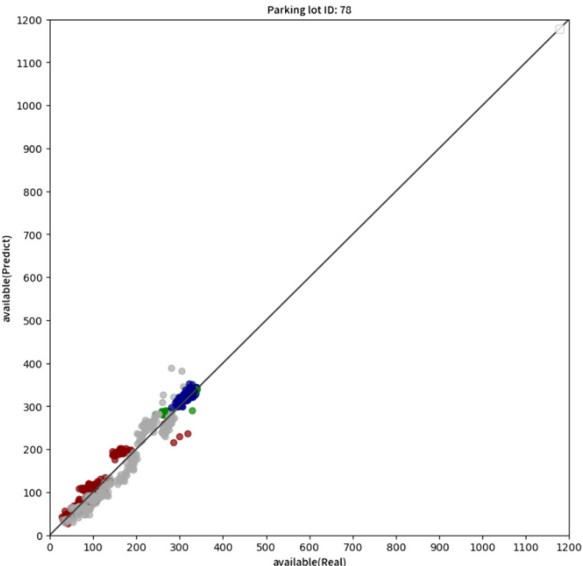

**Figure 6.** Prediction Results of Case 78.

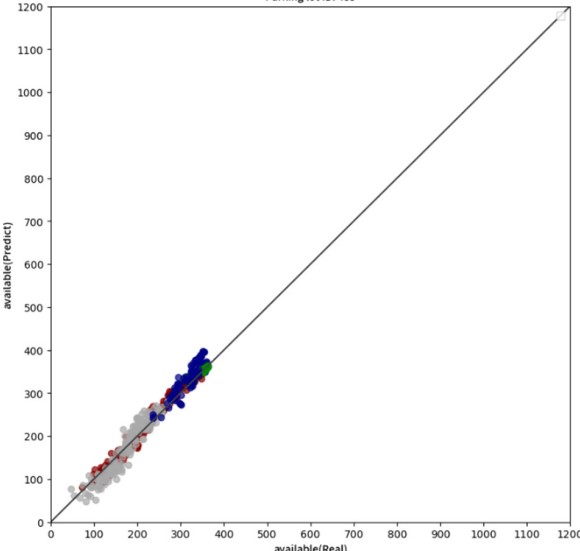

**Figure 7.** Prediction Results of Case 483.

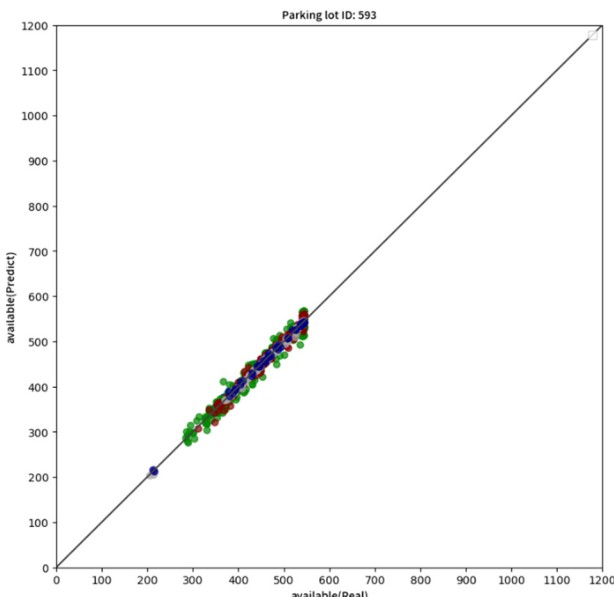

**Figure 8.** Prediction Results of Case 593.

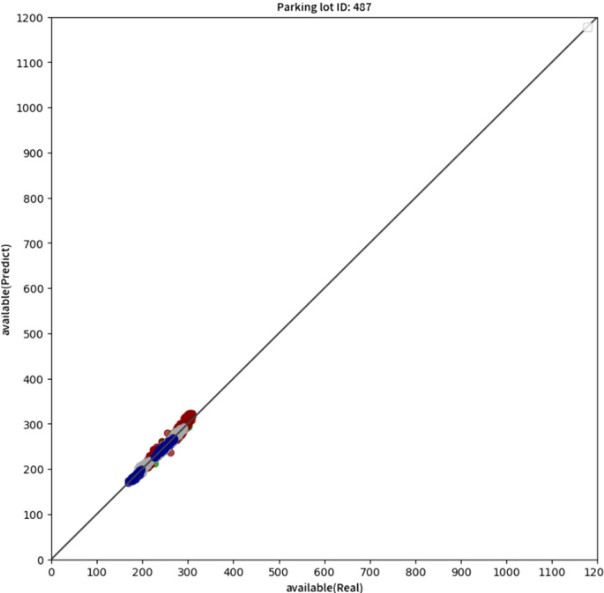

**Figure 9.** Prediction Results of Case 487.

Straight line:

Usually, this means that the parking lot may be disconnected at some point, and the number of disconnections will be fixed. In this case, the actual number will permanently be fixed since the number of available parking spaces will not change. However, since the prediction will still be estimated according to the normal situation, the calculated result will have an extensive variation range, so it is displayed as a straight line (case 299, Figure 10) on the graph. In addition, if the parking lot is only closed for a certain period in a day, and the number at the time of closing is a fixed value, a straight line still appears on the graph, but the points cluster more closely (case 585, Figure 11).

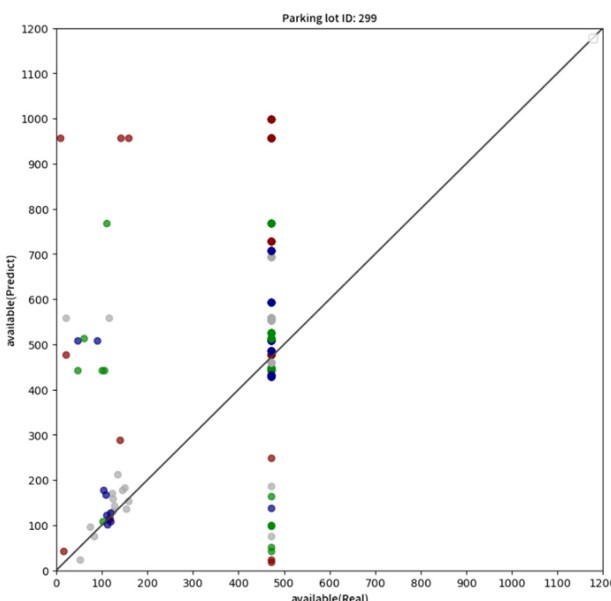

**Figure 10.** Prediction Results of Case 299.

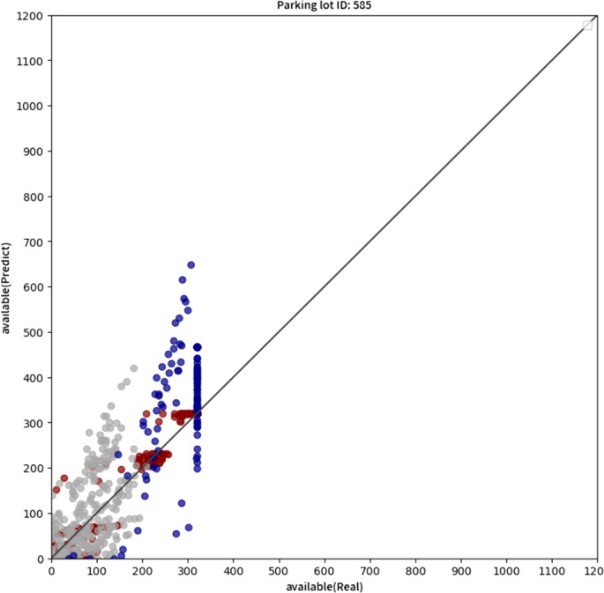

**Figure 11.** Prediction Results of Case 585.

Horizontal line:

Usually, this means that the parking lot may have a fixed period of closure and will be reopened after a while, and the number after the reopening can vary greatly depending on the day's conditions. Suppose the parking lot is open at a fixed time regardless of weekdays or holidays. Since the value at closing is almost always the same, the estimated value for the period after opening will also be the same. Since the data after the reopening are completely determined by the day's situation and cannot be the same forever (cases 650 and 377, Figures 12 and 13), they will result in a horizontal line. Another possible cause of this kind of graph is that the available number is always stable for some period in a day. The change is always significant in the next hour, resulting in multiple horizontal lines of different colors. This situation is usually only possible in large open parking lots (case 59, Figure 14). Generally speaking, it may be the result of system instability. When the system is frequently disconnected or has problems from time to time, such data performance will occur.

Mass of blocks:

Usually, this means that the parking lot has multiple time points in one day when the parking quantity changes significantly, and the parking quantity before this point falls within the daily stable range. Most of these kinds of parking lots are open 24 h. With the change in neighboring business districts, there may be an abnormal number of vehicles entering and leaving at various times throughout the day. The rectangular block below the line indicates underestimation (the real value is greater than the predicted value), which usually means that this period is the vehicle entry period. The rectangular block above the line indicates overestimation (the real value is less than the predicted value), usually meaning that this period is the vehicle departure period (cases 73, 15, Figures 15 and 16).

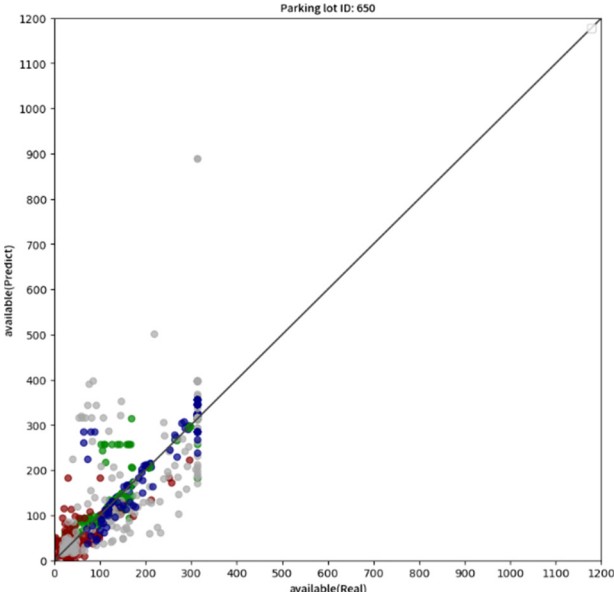

**Figure 12.** Prediction Results of Case 650.

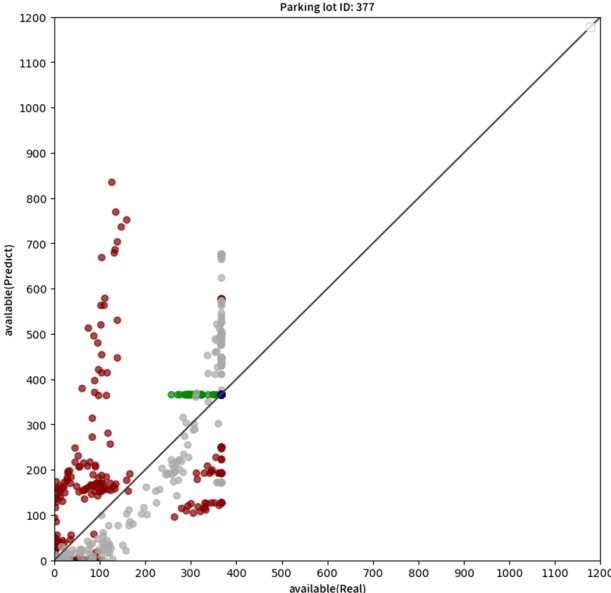

**Figure 13.** Prediction Results of Case 377.

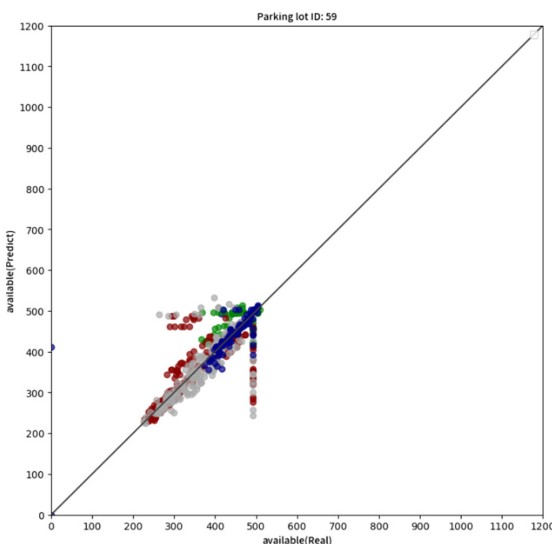

**Figure 14.** Prediction Results of Case 59.

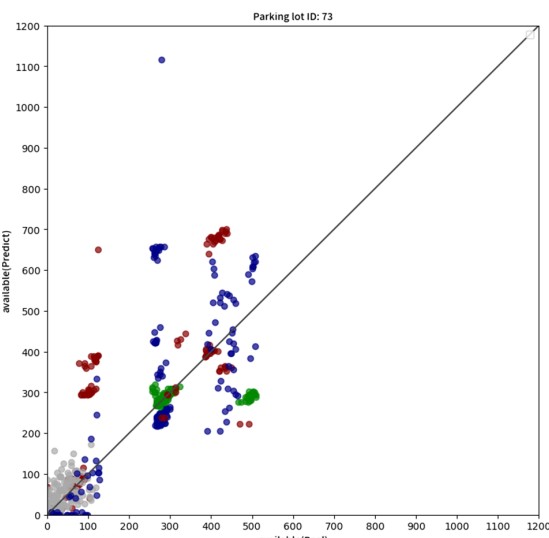

**Figure 15.** Prediction Results of Case 73.

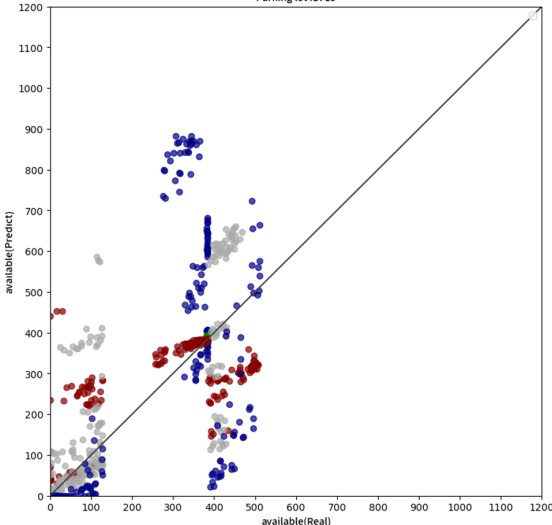

**Figure 16.** Prediction Results of Case 15.

From the above cases, we can find that abnormal data caused by communication or business control will inevitably exist in the parking data we want to observe. Such dirty data will inevitably have a significant impact on data modeling. Even with deep learning processing, the effect of dirty data cannot be eliminated. It has been observed that most of the previous studies used simple parking lot data for data verification so that acceptable prediction results could be obtained.

Restricted by resource constraints, it is impossible to continuously collect and calculate data for a large number of parking lots in a short period. Therefore, if the collected data are to be modeled and estimated in a short time, the frequency of data collection should not be too dense, and the computing performance should not be too heavy. Generally speaking, in practice, we cannot perform real-time data collection and complex modeling and prediction for a large number of parking lots simultaneously because they will require a lot of computing power to support.

We also used neural networks to compare predictions and observed Pearson correlation coefficients. From the results, it can be found that the prediction results are not good in most cases. In fact, because deep learning modeling requires a large amount of data, in the case of insufficient data, the results of each trained model are different. In this study, modeling training was repeated 20 times for each case, and the changes in the accuracy of these models were observed. As can be seen, under the same data usage, not only are the results very different, but also the accuracy cannot be effectively improved. This is not to say that deep learning is useless, but it should be used in a more suitable situation.

In this study, to collect and forecast data for a large number of parking lots simultaneously, the data collection frequency is one hour, and the rule-based modeling method is used for prediction and estimation to improve computing efficiency. In practice, we can also combine deep learning techniques with rule-based modeling methods and use this to achieve higher prediction accuracy. Since the optimized part is not the focus of this research, it will not be discussed in depth for now.

Evaluation of various dirty data is another issue that must be addressed. For example, a certain field may have fixed opening hours, or many vehicles may leave at the same time before closing, and the number of parking spaces used may fluctuate significantly at those times, or there may be machine equipment maintenance, which may cause abnormal transmission of various data. Furthermore, even the connection of some fields is unstable, which will cause the connection to be interrupted every few hours and the data to be abnormal, or, due to some particular factors, the field will be closed continuously for several weeks.

When these conditions occur, the data in this period will significantly impact the modeling. These scenarios will cause unpredictable non-linear changes in the available parking spaces in the parking lot and cause a substantial increase in the difficulty of estimation. Since the modeled data cannot be too messy, how to exclude the dirty data obtained from these conditions is another problem that can be investigated in-depth. Due to the complexity of this part of the work, it is not within the scope of this study.

For the above-mentioned particular scenarios with poor fitting results, other rules can be given to improve them. The critical point is the use of feature rules. Since the discussion of this part is beyond the scope of the core discussion of this study, it will not be discussed in depth here.

As enough data are often not accumulated in the early stage of field operations, the use of deep learning-based practices often faces the dilemma of data shortages. Less than a month of data are sufficient to make predictions if using the approach of this study, which can be a considerable advantage for the field used.

Since this study's focus is not on comparing accuracy, the accuracy results are not compared with other algorithms. In contrast, even if different deep learning model algorithms can achieve the same precision as in this study, the computational time and data required will significantly exceed the method used in this study.

Overall, using the approach proposed in this study can achieve satisfactory prediction results using less computing power and without relying on big data, compared to solutions that rely solely on deep learning.

## 6. Conclusions and Future Work

As a result of urban development, convenient transportation brings more vehicles on the road. However, the parking space is limited. We hope to encourage everyone to cherish the limited parking resources through price-based quantity. To fully integrate the parking resources of the whole city, perhaps the government can think about how to use the parking space more efficiently by dynamically changing the parking fee in an environment where all the shared parking lots are jointly operated. At the same time, according to the overall supply and demand in each small area, the standard of parking fees in each area can be planned.

When driving to a specific location, a driver usually relies on the directions of the parking navigation system to find a parking space. Generally speaking, drivers will inevitably choose a parking lot near the destination without certain price factors. However, if the price of parking in the nearest parking lot is too high, or if the parking lot is full, a driver will consider parking in a farther parking lot.

In such a situation, is it possible to develop a better regional joint pricing system to achieve vehicle evacuation in urban parking hotspots? Suppose we can predict the parking demand and supply in an area. In that case, we have the opportunity to formulate a reasonable price to affect the parking behavior in an area and then achieve the purpose of decentralized parking.

Consider a large area consisting of a group of adjacent areas. If the center point of this large area is a hot parking area, one can consider starting from the center point of the whole area and reducing the parking fee to the periphery of the area. As long as a certain proportion of vehicles in each small area move out to the periphery, there is a chance that the parking space in the center of the area will increase accordingly. Similarly, when it is desired to concentrate vehicles in a specific zone, it is only necessary to increase the parking fee in the adjacent areas outside the zone.

The following two diagrams are both composed of 25 small areas (Figure 17). The numbers in the figure represent the hourly parking fees in the area. As each area's price is different, drivers will usually park at nearby places with low prices. Therefore, with the joint control of regional prices, it will be possible to control the parking behavior in the region, trying to concentrate or move it, and thereby achieve the possibility of evacuating parking hotspots or creating emerging urban areas.

**Figure 17.** Demonstrations of the price distribution (standard areas).

However, in practical applications, since the price comes from demand, before setting the price of each area, one must know the supply and demand of each area. The total number of available parking spaces per hour will help design a regional joint pricing system and the dynamic parking fee mechanism can be used to control vehicle parking behavior more efficiently. The core idea of this research is also born from this point.

The situation in practice will be far more complicated than the model mentioned above. Since regions may be irregular and all regions are continuous, they may affect each other. The popularity of each region will be jointly determined by the chain change of all regions, even if its pricing has not changed.

In addition, some of these areas may experience sudden changes in parking usage within a short period. A change in a small area may affect the chain price changes in the peripheral area or even the entire area or cause abnormal price changes in the whole area over some time (as shown in Figure 18). Since this is already another in-depth question, it will be left as a developmental direction for follow-up research.

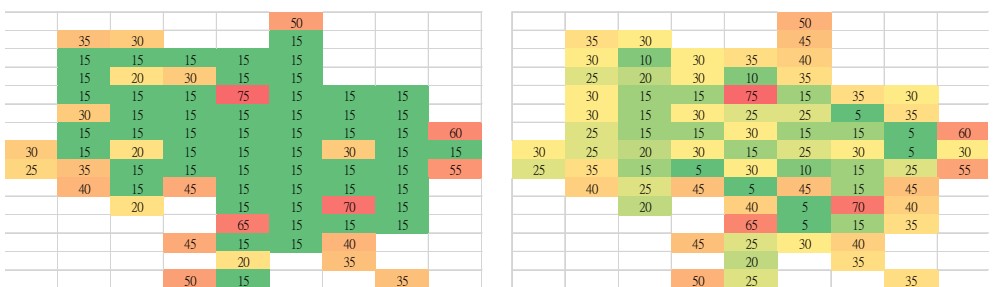

**Figure 18.** Demonstrations of the price distribution (irregular areas).

This research provides a fast and effective parking space estimation model, which can quickly calculate the area's total parking space supply capacity and this method can be used for subsequent system development.

The system can consider the drivers' backgrounds or other factors to estimate the reasonable hourly parking fee in the area. It can even further change the regional joint pricing to regional joint bidding to further achieve the purpose of parking space usage control.

The system will not only achieve the purpose of pricing by quantity but also change the mentality of drivers who expect to find a cheap parking space by chance. By directly throwing the decision of parking price into the hands of the market, the actual supply and demand of parking spaces determines the market price.

If it can cooperate with the automatic driving system, there may be more scheduling possibilities. Through the connection between the Internet of Vehicles system and the parking lot, the vehicle can drive to a nearby parking lot by itself and, when necessary, cooperate with the parking mechanism in the area to park in a suitable parking lot.

The current parking fee or the distance of the parking lot from the car owner can be used as consideration for selecting adjacent parking lots. The driver only needs to provide an estimated departure time, and the system can automatically complete all the intermediate matching processes.

The above is a brief description of the subsequent system development. In practical applications, shared parking spaces, car-hailing, ride-sharing, and autonomous driving systems can all be linked with the parking lot.

As shown in Figures 17 and 18, each grid in the figure is a specific area within the city. If the parking fee in some areas is high, the driver will inevitably park the vehicle in a nearby location with a lower fee (the green squares in the figures). With the popularization of the Internet of Vehicles and the improvement of network speed in the future, it will not be a problem to obtain the parking fees of all nearby locations. For price-sensitive driving, the parking location will also be changed under the influence of the parking price difference. The parking behavior and parking difficulty of the whole city can also be changed under the implementation of such a system. This is what this research refers to, the dynamic parking fee system designed for shared parking lots.

Such a system can involve both official and private parking operators. Of course, it is impossible for private operators alone to quickly grasp all parking information in the city. However, if the government provides suggested prices for private operators to follow up, it will be beneficial to both parties. Private operators can earn higher profits from it, and the

government can also relieve the parking pressure of parking hotspots with the assistance of private operators.

As long as the available parking spaces of each parking lot in the area can be quickly and accurately estimated in real time, the subsequent systems will be developed on this basis. Of course, the most important thing is that the 5G Internet of Vehicles in the city must be popularized. The software and hardware of important parking lots must be upgraded together so that there is a chance to make such a smart city shared parking dynamic pricing system a reality.

There are three main contributions of this study:

- This study provides a new idea to quickly estimate the current available parking spaces in a large area. This method can be implemented in the absence of data, which is very conducive to field application.
- This study sorts out the parking status of various fields, and discusses the possible real problems in various fields, as well as the possible impact on the estimation of available parking spaces.
- This study proposes a thinking strategy for using parking prices to control parking behavior at the city level, trying to quickly alleviate parking demand in hot areas with rapidly changing dynamic parking fees.

**Author Contributions:** Writing—review and editing, W.-M.C.; supervision, S.-M.W. All authors have read and agreed to the published version of the manuscript.

**Funding:** This research received no external funding.

**Institutional Review Board Statement:** Not applicable.

**Informed Consent Statement:** Not applicable.

**Data Availability Statement:** Data sharing not applicable.

**Conflicts of Interest:** The authors declare no conflict of interest.

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
