# Peer review of "Fast Way to Predict Parking Lots Availability: For Shared Parking Lots Based on Dynamic Parking Fee System"

_futureinternet, doi:10.3390/fi15030089_

Round 1

Reviewer 1 Report

Observations regarding article #2204311

General observations

-        The paper is well organized and proportionally structured

-        The introductory part is based on references that are not referred to in a more granular way – meaning that some ideas drawn from the study of literature is referred to in too large blocks of 2-4 works simultaneously, without nominating or specifying more in detail the achievements of these researches.

-        The study is compliant with general habits in the mentioned area (Taipei), with specific behavior of inhabitants – which may match, probably, several cities in the world with similar traffic characteristics and drivers’ behavior. However, it is possible that the applicability of the proposed approach might not suit different scenarios and countries.

-        The influence of weather conditions seems to be less considered. However, there are countries in Europe, and worldwide, where when raining or snowing, people avoid using public transportation in their commuting or traveling to work and back, using instead their own cars, due to commodity and not willing to walk in bad weather conditions. This aspect should be also taken into consideration, or at least observed in terms of influence on the mentioned “important rules”

Recommendations:

- Please try to improve the granularity of the introductory part in terms of references, giving more details on single cited papers, than of blocks of papers.

- Also try to add some information regarding the way the solution may be improved in future, and implemented on a larger scale, by reducing the work for calibration, and sharing information about the parking lots.

- The Conclusion section may also contain recommendations for industry sectors that may make use of your solution.

Author Response

Thanks for your time and comments, it’s really help for me to improve the paper.

Recommendations:

- Please try to improve the granularity of the introductory part in terms of references, giving more details on single cited papers, than of blocks of papers.

Response: I have rewritten the introduction section based on the references and provided more details about each single cited paper.

- Also try to add some information regarding the way the solution may be improved in future, and implemented on a larger scale, by reducing the work for calibration, and sharing information about the parking lots.

Response: I have added some details in the conclusion section about how to share and practice parking lot information on a large scale in the future.

- The Conclusion section may also contain recommendations for industry sectors that may make use of your solution.

Response: Same as above, I also pointed out some participating players and analyzed the respective benefits of the solution to the players and the government.

Reviewer 2 Report

Dear Authors, see the file attached

Author Response

Thanks for your time and comments, it’s really help for me to improve the paper.

1. Abstract – no comments, everything is ok, to my mind

2. Introduction:

a. There are too many normative statements, which are not proven by any documents. I advise to add some sources of information to all statements which can be supported by any kind of  documents. Introduction is not the part for expressing the opinions, everything should be  supported by evidences.

Response: I have rewritten the introduction section based on the references and provided more details about each single cited paper. I thought it would help to show more details.

b. I really lack the goal of the research. What for the authors do everything? It should be clearly stated in one sentence.

c. Next point – scientific value of the article, novelty. It should also be shown in short and precise form 

d. Practical value – the authors discuss the necessity for practical application, but do not show it in short summarised form.

Response: I have summarized the research contributions at the end of the article, I think this will help you understand the goal, novelty, and practical value of this research.

e. Limitations – there are some attempts to show the limitations of the study in Discussion section and in Conclusion, but again, these attempts are not summarised, very blurred, not  quite clear. 

Response: Since there are too many parts that can be improved, it is impossible to discuss all of them in this research. I try to describe some points that cannot be studied due to some reasons in the discussion section, and I hope that follow-up research can improve those parts.

  1. Related Work section – I like the content, but I do not understand the title of the section; the authors just show the stages of research and the obtained results. Moreover, the division of  locations is done, but it is not clear what is the basis for this division. I suppose, the authors  should describe the procedure of collecting information (“open data of the Taipei City  Government's parking”) and procedure and structure and participants of interviews (“obtained  some comments”) in Methods section.

Response: I have changed the title of this section to Current Study, perhaps this will be more in line with what is expressed here. As a well-known city in Asia, the state of parking information in Taipei should serve as a reference for the state of parking in most parts of Asia. The way to distinguish the types is mainly based on the parking behavior of the field. See the beginning of the Results section for information on data collection and use.

  1. There are many studies devoted to the parking issues; nevertheless, there is no section devoted to the study of previous researches in the area.

Response: I have added some of the previous research in the introductory section. Although this study discusses the parking problem, it mainly wants to alleviate the parking problem in the city through the price-quantity measure in a large area. Therefore, the main focus of the discussion is to quickly estimate the immediately available parking spaces, so there are many references to the estimation-related content, while the research on parking issues is relatively lacking.

  1. Methods section – the authors explain the modelling process, but they do not show all the steps usual for the scientific work dealing with statistical analysis, starting from stage of information collection, hypotheses formation, choice of soft for processing the information, and so on.

Moreover, the comment No.3 should be also considered. The Results section contains the material, the method of obtaining which is not shown either in Methods section.

Another point – the organization of formulae is strange; there are no numbers of formulae. And there appear the abbreviated form “???”, which should be “decoded”.

Response:

I added at the end of the modeling chapter when to use these modeling rules, we can use any statistical tool or programming language to implement these rules.

This study was validated using publicly available open data, which can be downloaded by anyone through the open data website of the Taipei City Government.

At the same time, I also gave a definition and a short description for avg in the content.

  1. Results section:
    a. Abbreviated forms should be written or described (tw97x, tw97y)
    b. Regression analysis – what are the criteria of model estimation? Why the authors take decision that 80% of the fields are valid? Why other 20% are not valid? The criteria must be applied. I suppose, there also should be some comments how to interpret the obtained results, it is Results section.
    c. Maybe, I am wrong, but I really do not see the necessity of Figure 4. Too big picture, and the interesting cases of it are demonstrated further. There is no any interpretation or explanation of the picture. Maybe, authors can consider the possibility to place this Figure in Appendix?

Response:
a. I have added the definitions of the abbreviations(tw97x, tw97y) to the content.

b. The result of the regression analysis here is just a numerical value. But if a parking system can accurately predict the demand of 80% of the field in practice, we believe that it will be of great help to improve the current parking situation or make profits in the future.

c. I added a brief description to the text. The Figure is for us to quickly see the distribution of prediction results of all cases. It can be found that in most cases, the predicted value is close to the actual value, so the points in the graph will move closer to the regression line. However, there are still a few cases where the graph exhibits anomalous scatter.

  1. Discussion section. This section usually contains comparison with other research, some disputable results, interesting cases, and so on. However, the authors place the results in this section. I think it is better to put the obtained results in the proper section. Another point refers to the content. In the title of the article the authors have “Dynamic Parking fee system”. There is no explanation/definition what it is. Then, in Discussion section the authors demonstrate the various lines referring to parking lots. Maybe, it is a good chance to “join together” the demandsupply of parking lots and fees for using them? The authors do not discuss the situation with fee and how to apply the dynamic fee system in the Results/Discussion section.

Response:

What I put in the Discussion section are indeed some rather special cases. It is easy to miss forecasts in these cases, and we discuss the reasons for this.

I have added some definitions and descriptions of dynamic parking fee systems in the future work in the concluding section. I also pointed out some participating players and analyzed the respective benefits of the solution to the players and the government.

  1. Conclusion. This section can comprise only the issues discussed in previous sections.Nevertheless, the authors place there the part of research/ results? If it is research – no explanation of methods. If it is results – from where these results were received? Then, Figure 17 – what for do the authors use it if they write that they are not going to discuss and research the issue?

Response:

I present the Results and Discussion in the Results and Discussion sections, respectively. In the Conclusions section, I present future work and research contributions.

Same as above, I have added some details about Fig. 17,18 in the Conclusions section about how to share and practice parking lot information on a large scale in the future.

Reviewer 3 Report

The manuscript presents an interesting method for estimating the total number of available parking spaces in some areas. The feasibility of the proposed method is demonstrated by some case studies.  Some specific comments are given in the following for the improvement of the manuscript. 

1. Lines 106-107 indicate "The focus of the study ... in the most efficient and general way."  But, the case studies are lack of the evidences to show that the proposed method is the most efficient and general in comparison with other existing schemes in the literature. 

2. In the Rule 2, it is unclear about how is the "Avg" actually executed?  Specifically, what is the length of the time period for "Avg"?

3.  Perhaps, it can be better understood and visualized if Rules 1-4 can be summarized by using some sorts of flow charts (block diagrams).  

4. It can be better understood if the equations of the regression analysis mentioned in Lines 345-350 can be provided and described.  

Author Response

Thanks for your time and comments, it’s really help for me to improve the paper.

  1. Lines 106-107 indicate "The focus of the study ... in the most efficient and general way." But, the case studies are lack of the evidences to show that the proposed method is the most efficient and general in comparison with other existing schemes in the literature.

Response: The results of prediction using neural networks have been supplemented in the Results section, and are also discussed briefly in Discuss Section.

  1. In the Rule 2, it is unclear about how is the "Avg" actually executed? Specifically, what is the length of the time period for "Avg"?

Response: I have added the definition of “Avg” to the text. The average referred to here is just a very simple average, and there is no special calculation process. For example, if we already know that there will only be 30% of the available parking spaces at 8:00 am compare to 7:00 am every day, and we know that there are 50 parking spaces available at 7:00 am one day, then the available parking spaces at 8:00 am that day can be inferred from this average.

Therefore, there may be 50x0.3=15 available parking spaces on that day. This average value is usually not affected by the conditions of the day and is always fixed. This study also uses this important feature to make predictions. As stated in the Results part, only about 1 to 2 months of data is enough to have a good prediction accuracy.

  1. Perhaps, it can be better understood and visualized if Rules 1-4 can be summarized by using some sorts of flow charts (block diagrams).

Response: I have added a flow chart of data processing at the end of the Modeling section, thank you for your suggestion.

  1. It can be better understood if the equations of the regression analysis mentioned in Lines 345-350 can be provided and described.

Response: Here we only use the coefficient of determination R2 of regression analysis to illustrate the correlation between the actual value and the predicted value, and no special regression equations is used. The larger R2 is, the better the prediction result is.

Round 2

Reviewer 2 Report

Thank you for all corrections. 

However, I still suppose all the apllied methods should be described, even if they are standard, and even if the collection of data is not the purpose of the research. It is so easy to write, for example, that the data obtained from the open sources and name these sources. 

Author Response

Thanks for your advice and help!

I have written the statistical method used at the end of Important Rule 4

“In fact, this is a chi-square goodness-of-fit test, mainly ...”

The chi-square goodness-of-fit test is a statistical method that can be used to test whether the samples are consistent. I use this method to distinguish whether the day belongs to a weekday or a holiday based on the data of a certain day.

I have added the open data download URL at the beginning of Sampling method

“From the open data of parking information in Taipei City (https://data.taipei/)...”

This website has a lot of public information about Taipei City that can be used for research, and it is completely free!

Reviewer 3 Report

The manuscript has been revised in accordance with the comments. It is presented in a better form and can be accepted for publication. 

Author Response

Thanks for your advice and help!